# Sex-Related Differences and Factors Associated with Peri-Procedural and 1 Year Mortality in Chronic Limb-Threatening Ischemia Patients from the CLIMATE Italian Registry

**DOI:** 10.3390/jpm13020316

**Published:** 2023-02-11

**Authors:** Eugenio Martelli, Matilde Zamboni, Giovanni Sotgiu, Laura Saderi, Massimo Federici, Giuseppe M. Sangiorgi, Mariangela V. Puci, Allegra R. Martelli, Teresa Messina, Paolo Frigatti, Maria Pia Borrelli, Carlo Ruotolo, Ilaria Ficarelli, Paolo Rubino, Francesco Pezzo, Luciano Carbonari, Andrea Angelini, Edoardo Galeazzi, Luca Calia Di Pinto, Franco M. Fiore, Armando Palmieri, Giorgio Ventoruzzo, Giulia Mazzitelli, Franco Ragni, Antonio Bozzani, Enzo Forliti, Claudio Castagno, Pietro Volpe, Mafalda Massara, Diego Moniaci, Elisa Pagliasso, Tania Peretti, Mauro Ferrari, Nicola Troisi, Piero Modugno, Maurizio Maiorano, Umberto M. Bracale, Marco Panagrosso, Mario Monaco, Giovanni Giordano, Giuseppe Natalicchio, Antonella Biello, Giovanni M. Celoria, Alessio Amico, Mauro Di Bartolo, Massimiliano Martelli, Roberta Munaó, Davide Razzano, Giovanni Colacchio, Francesco Bussetti, Gaetano Lanza, Antonio Cardini, Bartolomeo Di Benedetto, Mario De Laurentis, Maurizio Taurino, Pasqualino Sirignano, Pierluigi Cappiello, Andrea Esposito, Santi Trimarchi, Silvia Romagnoli, Andrea Padricelli, Giorgio Giudice, Adolfo Crinisio, Giovanni Di Nardo, Giuseppe Battaglia, Rosario Tringale, Salvatore De Vivo, Rita Compagna, Valerio S. Tolva, Ilenia D’Alessio, Ruggiero Curci, Simona Giovannetti, Giuseppe D’Arrigo, Giusi Basile, Dalmazio Frigerio, Gian Franco Veraldi, Luca Mezzetto, Arnaldo Ippoliti, Fabio M. Oddi, Alberto M. Settembrini

**Affiliations:** 1Department of General and Specialist Surgery Paride Stefanini, Faculty of Pharmacy and Medicine, Sapienza University of Rome, 155 viale del Policlinico, 00161 Rome, Italy; 2Medicine and Surgery School of Medicine, Saint Camillus International University of Health Sciences, 8 via di Sant’Alessandro, 00131 Rome, Italy; 3Division of Vascular Surgery, Department of Cardiovascular Sciences, S. Anna and S. Sebastiano Hospital, via F. Palasciano, 81100 Caserta, Italy; 4Division of Vascular Surgery, Saint Martin Hospital, 22 viale Europa, 32100 Belluno, Italy; 5Clinical Epidemiology and Medical Statistics Unit, Department of Medicine, Surgery and Pharmacy, University of Sassari, viale San Pietro, 07100 Sassari, Italy; 6Department of Systems Medicine, University of Rome Tor Vergata, 1 viale Montpellier, 00133 Rome, Italy; 7Department of Biomedicine and Prevention, University of Rome Tor Vergata, 1 viale Montpellier, 00133 Rome, Italy; 8Medicine and Surgery School of Medicine, Campus Bio-Medico University of Rome, 21 via À. del Portillo, 00128 Rome, Italy; 9Division of Anesthesia and Intensive Care of Organ Transplants, Umberto I Polyclinic University Hospital, 155 viale del Policlinico, 00161 Rome, Italy; 10Divisions of Vascular Surgery, S. Maria Misericordia University Hospital, 15 Piazzale Santa Maria della Misericordia, 33100 Udine, Italy; 11Divisions of Vascular Surgery, Cardarelli Hospital, 9 Via A. Cardarelli, 80131 Naples, Italy; 12Divisions of Vascular Surgery, Pugliese Ciaccio Hospital, 83 viale Pio X, 88100 Catanzaro, Italy; 13Divisions of Vascular Surgery, Riuniti University Hospitals, 71 via Conca, Torrette (AN), 60126 Ancona, Italy; 14Divisions of Vascular Surgery, Treviso Hospital, 1 piazzale del’Ospedale, 31100 Treviso, Italy; 15Divisions of Vascular Surgery, SS. Annunziata Hospital, 31 via dei Vestini, 66100 Chieti, Italy; 16Divisions of Vascular Surgery, San Donato Hospital, 20 via Pietro Nenni, 52100 Arezzo, Italy; 17Divisions of Vascular Surgery, San Matteo Polyclinic, 19 viale Camillo Golgi, 27100 Pavia, Italy; 18Divisions of Vascular Surgery, Infermi Hospital, Via dei Ponderanesi 2, 13875 Ponderano, Italy; 19Divisions of Vascular Surgery, Bianchi-Melacrino-Morelli Hospital, 21 via G. Melacrino, 89124 Reggio di Calabria, Italy; 20Divisions of Vascular Surgery, San Giovanni Bosco Hospital, 3 piazza del Donatore di Sangue, 10154 Turin, Italy; 21Divisions of Vascular Surgery, Department of Translational Research and New Technologies in Medicine and Surgery, University of Pisa, 2 via Paradisa, 56124 Pisa, Italy; 22Divisions of Vascular Surgery, Gemelli Molise Hospital, 1 largo A. Gemelli, 86100 Campobasso, Italy; 23Divisions of Vascular Surgery, Federico II Polyclinic, Department of Public Health and Residency Program in Vascular Surgery, University of Naples Federico II, 5 via S. Pansini, 80131 Naples, Italy; 24Divisions of Vascular Surgery, Pineta Grande Hospital, Km. 30 via Domitiana, 81030 Castelvolturno, Italy; 25Divisions of Vascular Surgery, Sanatrix Clinic, 31 via S. Domenico, 80127 Naples, Italy; 26Divisions of Vascular Surgery, Venere Hospital, 1 via Ospedale di Venere, 70131 Bari, Italy; 27Divisions of Vascular Surgery, Sant’Andrea Hospital, 197 via Vittorio Veneto, 19121 La Spezia, Italy; 28Divisions of Vascular Surgery, MultiMedica Hospital, 300 via Milenese, 20099 Sesto San Giovanni, Italy; 29Divisions of Vascular Surgery, San Pio Hospital, 1 via dell’angelo, 82100 Benevento, Italy; 30Divisions of Vascular Surgery, F.Miulli Hospital, Strada Prov. 127 Acquaviva-Santeramo Km. 4, 70021 Acquaviva delle Fonti, Italy; 31Divisions of Vascular Surgery, Multimedica Hospital, 70 viale Piemonte, 21053 Castellanza, Italy; 32Divisions of Vascular Surgery, Monaldi Hospital, via L. Bianchi, 84100 Naples, Italy; 33Divisions of Vascular Surgery, Department of Molecular and Clinical Medicine, Sapienza University of Rome, Giorgio Nicola Papanicolau, 00189 Rome, Italy; 34Divisions of Vascular Surgery, Sant’Andrea University Hospital, 1035/1039 via di Grottarossa, 00189 Rome, Italy; 35Divisions of Vascular Surgery, San Carlo Hospital, via Potito Petrone, 85100 Potenza, Italy; 36Divisions of Vascular Surgery, Department of Clinical and Community Sciences, University of Milan, 19 via della Commenda, 20122 Milan, Italy; 37Divisions of Vascular Surgery, Maggiore Polyclinic Hospital Ca’ Granda IRCCS and Foundation, 35 via Francesco Sforza, 20122 Milan, Italy; 38Divisions of Vascular Surgery, Salus Clinic, 4 via F. Confalonieri, 84091 Battipaglia, Italy; 39Divisions of Vascular Surgery, San Marco Hospital, viale Carlo Azeglio Ciampi, 95121 Catania, Italy; 40Divisions of Vascular Surgery, Pellegrini Hospital, 41 via Portamedina alla Pignasecca, 80134 Naples, Italy; 41Divisions of Vascular Surgery, Niguarda Hospital, Piazza dell’Ospedale Maggiore 3, 20161 Milan, Italy; 42Divisions of Vascular Surgery, Maggiore Hospital, 10 Piazza Ospitale, 26900 Lodi, Italy; 43Divisions of Vascular Surgery, Garibaldi-Nesima Hospital, 636 via Palermo, 95122 Catania, Italy; 44Divisions of Vascular Surgery, Vimercate Hospital, 10 via Cosma e Damiano, 20871 Vimercate, Italy; 45Divisions of Vascular Surgery, University Hospital Pietro Confortini, 1 Piazzale Aristide Stefani, 37126 Verona, Italy

**Keywords:** chronic limb-threatening ischemia, outcome, sex, age, limb salvage

## Abstract

Background: Identifying sex-related differences/variables associated with 30 day/1 year mortality in patients with chronic limb-threatening ischemia (CLTI). Methods: Multicenter/retrospective/observational study. A database was sent to all the Italian vascular surgeries to collect all the patients operated on for CLTI in 2019. Acute lower-limb ischemia and neuropathic-diabetic foot are not included. Follow-up: One year. Data on demographics/comorbidities, treatments/outcomes, and 30 day/1 year mortality were investigated. Results: Information on 2399 cases (69.8% men) from 36/143 (25.2%) centers. Median (IQR) age: 73 (66–80) and 79 (71–85) years for men/women, respectively (*p* < 0.0001). Women were more likely to be over 75 (63.2% vs. 40.1%, *p* = 0.0001). More men smokers (73.7% vs. 42.2%, *p* < 0.0001), are on hemodialysis (10.1% vs. 6.7%, *p* = 0.006), affected by diabetes (61.9% vs. 52.8%, *p* < 0.0001), dyslipidemia (69.3% vs. 61.3%, *p* < 0.0001), hypertension (91.8% vs. 88.5%, *p* = 0.011), coronaropathy (43.9% vs. 29.4%, *p* < 0.0001), bronchopneumopathy (37.1% vs. 25.6%, *p* < 0.0001), underwent more open/hybrid surgeries (37.9% vs. 28.8%, *p* < 0.0001), and minor amputations (22% vs. 13.7%, *p* < 0.0001). More women underwent endovascular revascularizations (61.6% vs. 55.2%, *p* = 0.004), major amputations (9.6% vs. 6.9%, *p* = 0.024), and obtained limb-salvage if with limited gangrene (50.8% vs. 44.9%, *p* = 0.017). Age > 75 (HR = 3.63, *p* = 0.003) is associated with 30 day mortality. Age > 75 (HR = 2.14, *p* < 0.0001), nephropathy (HR = 1.54, *p* < 0.0001), coronaropathy (HR = 1.26, *p* = 0.036), and infection/necrosis of the foot (dry, HR = 1.42, *p* = 0.040; wet, HR = 2.04, *p* < 0.0001) are associated with 1 year mortality. No sex-linked difference in mortality statistics. Conclusion: Women exhibit fewer comorbidities but are struck by CLTI when over 75, a factor associated with short- and mid-term mortality, explaining why mortality does not statistically differ between the sexes.

## 1. Introduction

Chronic limb-threatening ischemia (CLTI) affects 1–10% of patients with peripheral arterial disease (PAD), represents its advanced stages, and is characterized by rest pain or tissue necrosis in the foot. It generally results from involvement of at least two arterial segments (aorto-iliac, femoropopliteal, and tibiopedal) or severe tibiopedal disease. The latter is particularly involved in patients with diabetes mellitus, end-stage renal disease, or the very elderly. It represents a very broad range of severe malperfusion of the lower limb and the associated limb threat. The general and limb prognosis of these patients is adverse: they are at continuous risk of a major cardiovascular event, sudden death, and, of course, amputation [1].

The survival of female and male patients who underwent treatment for CLTI has been investigated with discordant results. For instance, some studies from Germany and the USA showed that 30 day mortality was significantly higher in females, while others from Japan and Sweden, respectively, identified female sex as a significant positive predictor of 2 year survival or reported that male sex was significantly associated with an increased risk of death [2,3,4,5,6,7,8,9,10,11].

Understanding the pathophysiological differences between the sexes is important to improving the quality of care. In this setting, it is reported that a lower rate of diagnostic angiograms and interventional procedures are performed in women compared with men [12]. This has raised the concern that the therapeutic approach to cardiovascular diseases should be sex-specific because of the existence of sex-related disparities in cardiovascular physiology [13,14]. Sex differences have been identified as additional determinants in diagnostic definitions and referral requirements for some diseases and sex-specific treatments are set including percutaneous coronary intervention, coronary artery bypass graft surgery, and PAD [15]. In particular, factors such as older age, late presentation, delayed diagnosis, smaller-size vessels, and other sex-related biases have been postulated to account, at least in part, for the portended less-favorable outcome in women with PAD. In addition, most studies on PAD have had low enrollment rates for women. Fortunately, the sex disparity in the management of PAD has been recognized, and more effort and resources have been dedicated to studying this issue. Men and women have distinct and significant biological differences. Physiologically, women differ from men in many respects (e.g., they have smaller blood vessels; their menopausal state and eventual estrogen replacement therapy can affect their cardiovascular risk; etc.) [16,17,18]. It is possible that these differences may contribute to the different presentations of the disease between the sexes and the postoperative complications of major vascular procedures.

Related to the increased awareness of sex differences, the objective of the present study is to evaluate sex-related differences in the immediate post-surgery outcome and 1 year mortality in patients affected by CLTI. Secondary endpoints consist of the identification of any demographic, risk factor for atherosclerosis, comorbidity, or treatment significantly associated with operative and 1 year mortality.

## 2. Materials and Methods

CLIMATE (Chronic Limb-threatening Ischemia Mortality At short-medium Term and Sex) is a multicenter and retrospective observational study.

The same ad hoc electronic questionnaire was sent by email to all 143 Italian Divisions of Vascular Surgery, which consist of 20 (14% academic) and 123 (86% non-academic) centers. The questionnaire asked to anonymously collect data regarding all the patients treated in each center from 1 January to 31 December 2019 for the first episode of CLTI on the target limb (by endovascular, surgical, hybrid revascularization, regenerative cellular therapy, or major amputation). Patients with acute lower limb ischemia or exclusively neuropathic diabetic foot (non-ischemic, i.e., with triphasic wave distal arterial blood-flow at duplex scan) were not an object of this study and were not considered in the database. Follow-up was limited to the first year after the operative treatment.

Data on demographics, risk factors for atherosclerosis, comorbidities, clinical presentation, treatment, technical and clinical success, postoperative medical therapy, limb salvage, 30 day and 1 year mortality, and cause of death were collected from clinical charts, operator reports, discharge letters, institutions’ archives, and reported on the electronic database by each division of vascular surgery. The result from each variable in the database was classified as reported in Table 1 (for instance, 0 = no, 1 = yes), that is, ready for statistical analysis. Each center provided two surgeons for this study: one deputed to data collection and the other the chief of the division, who was responsible for the accuracy and integrity of the data. All the 35 databases were checked for congruency and summarized together in the original database from the first author (E.M.). Further supervision of all the data collected from all the centers was performed by the co-first author (M.Z.) and by the statistician co-authors #3, 4, and 7 (G.S., L.S., and M.P.). Data supporting the findings of this study are available from the corresponding author upon reasonable request. Weekly web meetings were held between the authors and the study group while drafting the protocol and over the following two months of patient recruitment, to standardize data collection.

Here are some definitions we adopted; the remaining are reported in the tables. Hyperlipemia: low-density lipoprotein ≥ 70 mg/dL, total cholesterol > 135 mg/dL, or triglyceride > 150 mg/dL. Arterial hypertension is defined as having a systolic and/or diastolic blood pressure of ≥140 mmHg and ≥90 mmHg, respectively. Coronary artery disease (CAD): stable or unstable angina, ejection fraction < 30%, history of myocardial infarction or congestive heart failure. Chronic obstructive pulmonary disease (COPD): symptomatic, but sometimes with only radiological signs. Cerebrovascular disease (CVD): previous TIA or stroke. Minor amputation: toe or trans-metatarsal amputation. Limb salvage (LS): any treatment for CLTI that is successful in avoiding a major amputation. Use of antiplatelets, anticoagulants, and statins was classified as monotherapy, two medications, or three or more medications.

Institutional review board approval and patient-informed consent were waived. The current Italian legislation on observational studies (our study falls under this category) does not request the above-mentioned documents when clinical data are anonymized (Official Gazette of the Italian Republic #76, 31 March 2008).

### Statistical Analysis

Sample characteristics were collected in an ad hoc dataset (Appendix A).

Qualitative variables were summarized with absolute and relative (percentage) frequencies; quantitative ones with medians and interquartile ranges (IQR). Pearson’s or Fisher’s exact tests were used to evaluate differences in qualitative variables between males and females, whereas the Mann–Whitney test was performed to compare quantitative variables.

Survival analysis at 30 days and at 1 year was performed by Cox proportional hazard regression. Candidate variables for multivariate analysis were chosen if they were statistically significant at univariate analysis or clinically relevant.

The Kaplan–Meier curve and log-rank tests were performed to describe survival according to gender.

A *p*-value of less than 0.05 was considered statistically significant. The STATA13 statistical software was used for all statistical computations. 

## 3. Results

Thirty-six (25.2%) of the 143 divisions of vascular surgery from 17 of the 20 Italian regions replied to the invitation and joined the study. The proportion between the typology of the adhering centers, compared to the typology of the Italian Divisions of Vascular Surgery, was found to be constant: five (14.3%) academic vs. thirty-five (85.6%) non-academic. Information on 2399 cases was collected. All the data requested from the database was obtained from each vascular surgery center, and at follow-up, no patient was lost at 30 days, while 20 (0.8%) were missing at 1 year.

Table 1 shows the sample characteristics stratified by sex.

**Table 1 jpm-13-00316-t001:** Sample characteristics.

Variables		Total Cohort (n = 2399)	Men (n = 1677)	Women (n = 722)	*p*-Value
Median (IQR) age, yrs		75 (67–81)	73 (66-80)	79 (71–85)	<0.0001
Age > 75 yrs		1128 (47.0)	672 (40.1)	456 (63.2)	0.0001
Tobacco use:		1538 (64.2)	1234 (73.7)	304 (42.2)	<0.0001
	never	857 (35.8)	441 (26.3)	416 (57.8)	<0.0001
	former (stop > 10 yrs)	840 (35.1)	693 (41.4)	147 (20.4)	<0.0001
	smoker	698 (29.1)	541 (32.3)	157 (21.8)	<0.0001
Overt diabetes mellitus (yes vs. no):		1418 (59.2)	1038 (61.9)	380 (52.8)	<0.0001
	no	978 (40.8)	638 (38.1)	340 (47.2)	<0.0001
	non-insulin dependent	717 (29.9)	529 (31.6)	188 (26.1)	0.007
	insulin dependent	701 (29.3)	509 (30.4)	192 (26.7)	0.068
Hyperlipemia:		1601(66.9)	1160 (69.3)	441 (61.3)	<0.0001
	no	794 (33.1)	515 (30.8)	279 (38.6)	0.001
	under therapy	1498 (62.6)	1089 (65.0)	409 (56.8)	0.0001
	no therapy	103 (4.3)	71 (4.2)	32 (4.4)	0.825
Arterial hypertension:		2176 (90.8)	1538 (91.8)	638 (88.5)	0.011
	no	221 (9.2)	138 (8.2)	83 (11.5)	0.011
	under therapy	2147 (89.6)	1515 (90.4)	632 (87.7)	0.047
	no therapy	29 (1.2)	23 (1.4)	6 (0.8)	0.245
Chronic renal insufficiency:		626 (26.1)	453 (27.0)	173 (24.0)	0.123
	no	1769 (73.9)	1222 (73.0)	547 (76.0)	0.125
	creatinine > 2 mg/dL	408 (17.0)	283 (16.9)	125 (17.4)	0.765
	hemodialysis treatment	218 (9.1)	170 (10.1)	48 (6.7)	0.006
Coronary artery disease:		947 (39.6)	736 (43.9)	211 (29.4)	<0.0001
	no	1447 (60.4)	939 (56.0)	508 (70.7)	<0.0001
	revascularized	705 (29.5)	559 (33.4)	146 (20.3)	<0.0001
	non-revascularized	242 (10.1)	177 (10.6)	65 (9.0)	0.246
Chronic obstructive pulmonary disease:		806 (33.7)	622 (37.1)	184 (25.6)	<0.0001
	no	1589 (66.3)	1054 (62.9)	535 (74.4)	<0.0001
	only radiological signs	479 (20.0)	363 (21.7)	116 (16.1)	0.002
	symptomatic	327 (13.7)	259 (15.4)	68 (9.5)	0.0001
Cerebrovascular disease:		186 (7.8)	128 (7.6)	58 (8.0)	0.736
	no	2213 (92.2)	1549 (92.4)	664 (92.0)	0.918
	previous TIA	139 (5.8)	95 (5.7)	44 (6.1)	
	previous stroke	47 (2.0)	33 (2.0)	14 (1.9)	
Rutherford category:	4 (rest pain)	964 (40.2)	697 (41.6)	267 (37.0)	0.107
	5 (minor tissue loss)	1078 (44.9)	738 (44.0)	340 (47.1)	
	6 (major tissue loss)	357 (14.9)	242 (14.4)	115 (15.9)	
Necrosis/infection of the foot:	no	1194 (49.9)	854 (51.0)	340 (47.4)	0.202
	dry	603 (25.2)	407 (24.3)	196 (27.3)	
	wet	596 (24.9)	414 (24.7)	182 (25.4)	
First intervention:	endovascular only (rarely, regenerative cellular therapy)	1366 (57.1)	922 (55.2)	444(61.6)	0.004
	any open revascularization surgery	840 (35.2)	632 (37.9)	208 (28.8)	<0.0001
	any major amputation	184 (7.7)	115 (6.9)	69 (9.6)	0.024
Any intervention below the knee		1287 (53.9)	912 (54.6)	375 (52.3)	0.313
Technical success of CLTI revascularization:	no	211 (9.5)	152 (9.7)	59 (9.0)	0.611
	yes	2004 (90.5)	1410 (90.3)	594 (91.0)	
Associated minor amputation		467 (19.5)	368 (22.0)	99 (13.7)	<0.0001
Postoperative antiplatelets, anticoagulants, statins:	monotherapy	561 (24.3)	373 (23.1)	188 (26.9)	0.105
	two medications	1229 (53.2)	863 (53.6)	366 (52.4)	
	three or more medications	519 (22.5)	375 (23.3)	144 (20.6)	
Clinical success of CLTI revascularization:	worsen	157 (7.1)	109 (7.0)	48 (7.4)	0.951
	no change	343 (15.5)	142 (15.5)	101 (15.5)	
	improved	1716 (77.4)	1212 (77.5)	504 (77.2)	
Limb salvage:		1965 (82.1)	1382 (82.6)	583 (80.9)	0.307
	in Rutherford category 4	882 (44.9)	638 (46.2)	244 (41.9)	0.080
	in Rutherford category 5	917 (46.7)	621 (44.9)	296 (50.8)	0.017
	in Rutherford category 6	166 (8.5)	123 (8.9)	43 (7.4)	0.275
30 day mortality		74 (3.1)	44 (2.6)	30 (4.2)	0.047
1 year mortality		317 (13.5)	211 (12.8)	106 (14.9)	0.167
Cause of death:	cardiac	141 (42.1)	91 (41.2)	50 (43.9)	0.635
	neurologic	19 (5.7)	10 (4.5)	9 (7.9)	0.202
	pulmonary	33 (9.9)	31 (14.0)	3 (2.6)	0.001
	cancer	19 (5.7)	13 (5.9)	6 (5.3)	0.822
	multi-organ failure	46 (13.7)	25 (11.3)	21 (12.4)	0.074
	other	77 (23.0)	51 (23.1)	25 (21.9)	0.804

Quantitative variables are expressed with a median and interquartile range (IQR), and qualitative ones as absolute and relative (percentage) frequencies, n (%). TIA, or transient ischemic attack. CLTI, chronic limb-threatening ischemia.

Among 2399 patients, 1677 (69.9%) were males; the median (IQR) age in the sample was 75 (67–81) years old, with a significant difference between men and women [73 (66–80) years vs. 79 (71–85) years, *p* < 0.0001, respectively]. Women were older than 75 compared to men (63.2% vs. 40.1%, *p* = 0.0001). The age 75 cutoff (≤75 or >75) was chosen on the basis of the median value.

The most common cardiovascular risk factor and comorbidity in the total cohort are, respectively, arterial hypertension and CAD, followed by hyperlipemia, tobacco use, diabetes mellitus, COPD, chronic renal insufficiency, and CVD.

A significantly greater proportion of men who were smokers and affected by diabetes, dyslipidemia, end-stage renal disease on hemodialysis treatment, arterial hypertension, CAD and COPD underwent significantly more open or hybrid surgeries for CLTI revascularization, and amputations of the toes or the forefoot as a complementary treatment for LS. On the contrary, a significantly greater proportion of women underwent less-invasive direct or indirect revascularizations for CLTI, mainly endovascular (cellular therapy was used in only 18, 2.5% of women, and 27, 1.6% of men) and major amputations.

CVD, the clinical presentation according to Rutherford’s classification, infection/necrosis of the foot, above the knee vs. below the knee revascularization, technical and clinical success of revascularization for CLTI, postoperative medical therapy, 1 year mortality, cardiac, neurologic, malignant, and multi-organ failure causes of death do not statistically differ between the sexes, as well as 30 day mortality, which, despite being close to being so, is not statistically significant (*p* = 0.047, which is approximately *p* = 0.05). Instead, significantly more women with limited tissue loss (Rutherford category 5) obtain LS, and significantly more males die from pulmonary causes.

Table 2 reports the Cox regression analysis to assess the relationship between demographic, epidemiological, and clinical characteristics, and 30 day mortality.

Results from multivariate analysis show that age > 75 years was the only variable associated with the 30 day mortality rate (HR = 3.69, 95%CI: 1.53–8.62; *p* = 0.003).

Table 3 shows the results of a Cox regression analysis to assess the relationship between demographic, epidemiological, clinical characteristics, and 1 year mortality.

Results from multivariate analysis show that the following factors were found to be associated with 1 year mortality: age > 75 years (HR = 2.14, 95%CI: 1.60–2.87; *p* < 0.0001); therapy for hyperlipemia (HR = 0.69, 95%CI: 0.51–0.93; *p* = 0.015); chronic renal insufficiency (CRI) (HR = 1.54, 95%CI: 1.28–1.84; *p* < 0.0001); CAD (HR = 1.26, 95%CI: 1.02–1.57; *p* = 0.036); dry necrosis (HR = 1.42, 95%CI: 1.02–1.98; *p* = 0.040); and wet necrosis (HR = 2.04, 95%CI: 1.46–2.85; *p* < 0.0001) of the foot.

Figure 1 and Figure 2 show the Kaplan–Meier survival curves (overall distribution and for men/women) at 30 days and 1 year, respectively. Again, no statistically significant difference is noted between the sexes.

## 4. Discussion

The aim of the study was to evaluate the differences between men and women in 30 days and 1 year mortality. Our findings showed that no differences were observed between the sexes.

The overall 30 day mortality rate in this study of 2399 CLTI patients is 3.1%. One-year mortality is 13.5%, which is much lower than the 23.1–28.7% reported in the COPART French registry on 411 patients treated for CLTI [19]. Differences in risk factors and comorbidities between the French and Italian populations may explain the difference in 1 year mortality between these two studies.

Notwithstanding that our male patients were significantly more smokers, affected by diabetes, end-stage renal disease on hemodialysis treatment, arterial hypertension, CAD, and COPD, and revascularized for CLTI by open surgery, they were also significantly younger, more on statin therapy, and more revascularized for CAD, potentially inducing more protection from cardiovascular risk. 

Additionally, the male patients in the Medicare population affected by CLTI and analyzed from 2015 to 2017 received significantly more statin therapy compared to females [4].

The current clinical practice guidelines from the European Society for Vascular Surgery strongly recommend the use of moderate or high-intensity statin therapy to reduce all-cause and cardiovascular mortality in patients with CLTI [20].

Cardiologists from the University of Minnesota found that patients with unprotected left main coronary artery disease benefit from preoperative coronary artery revascularization before vascular surgery [21].

Our findings that some vascular beds are more affected than others when comparing men and women (i.e., the coronary district vs. the peripheral arterial district) and that some risk factors are so disparate between the groups suggest a potentially different phenotypic expression of the disease process. However, it is important to notice that this result might merely be a function of which men and women were selected for the treatment.

Anesthesiologists from Melbourne, Toronto, and Auckland have recently evaluated the effects of randomized interventions by sex in large international perioperative trials and concluded that women were healthier than men, but outcomes were similar. These authors encouraged further research to understand the reason for this discrepancy [22].

Colleagues from Auckland and Hamilton analyzed 1773 patients with CLTI in the midland region of New Zealand over a 12 year period. They found a worse long-term survival rate for women with CLTI, despite 30 day mortality not differing depending on sex [23].

A recent German study from the cardiologists at the University of Muenster on almost 200,000 unselected patients treated for CLTI over an 8 year period showed that 30 day mortality is significantly higher in women [3]. The same results were obtained from the US National Inpatient Sample database [7,10].

Other French colleagues from the University of Strasbourg demonstrated a significantly lower survival rate at 6 years, but not at 30 days, among women compared to men undergoing infra-inguinal open surgery for CLTI. They concluded that female sex was an independent factor predicting death [24].

A possible explanation for this worse long-term survival in women treated for CLTI is that they are associated with more severe disease at presentation (although in our study Rutherford categories were similar in both sexes), develop arteriosclerotic changes later in life, and require treatment at an older age [16].

On the contrary, a recent multicenter study from Japan has identified female sex as a significant positive predictor of 2 year overall survival in patients treated for CLTI [11].

Another Swedish population-based study conducted between 2008 and 2013 on over 10,000 patients who underwent revascularization for CLTI and were followed up for a median of 2.7 years reported that male sex was significantly associated with an increased risk of amputation or death at multivariate analysis [6].

This dichotomy could open a reflection on the genetic, environmental, and dietary factors implied on the outcome of CLTI patients. The Atherosclerosis Risk in Communities (ARIC) study has already focused attention on the association of race (“Blacks vs. Whites” in the ARIC study) with incident CLTI-related hospitalizations that leads to differences in clinical disease risk and presentation [25].

We have found advanced age over 75 years old to be a negative prognostic factor, both for 30 day and 1 year mortality. A Dutch study recently reported a similar result for 1 year mortality in advanced age, as did a Yale School of Medicine study [26,27].

The older population is increasing, and this knowledge of worse CLTI outcomes for the elderly population is important for clinical decision making.

Our female patients treated for CLTI are significantly older compared to males, and this is in contrast with the exception of the Italian data (females are 0.9 years younger than males) reported in the VASCUNET and International Consortium of Vascular Registries [28].

Our analysis demonstrates that females are struck by CLTI at an age >75 years, a pivotal factor associated with short- and mid-term mortality. This fact may explain why the mortality rate does not significantly differ between the sexes, although females have fewer risk factors and comorbidities associated with this condition.

Our study confirms other independent predictive factors for mortality at 1 year, namely CRI, CAD, and tissue loss.

CLTI is a terminal manifestation of systemic atherosclerosis. Therefore, it is often accompanied by clinically significant CAD, resulting in high mortality. The goal of treating patients with CLTI is not only to save a functional limb but also to improve cardiovascular outcomes. While some risk factors (age and sex) are immutable, others are (cigarette smoking, dyslipidemia, diabetes mellitus, a sedentary lifestyle, and treatable hypertension). In the absence of an efficient cardiovascular work-up and aggressive treatment of risk factors and associated comorbidities, the prognosis of CLTI is generally poor [29,30,31].

End-stage renal disease and tissue loss are established critical factors for mid-term mortality in patients undergoing revascularization for CLTI [32,33,34,35].

Interestingly, in our study, although significantly more women underwent major amputations, LS was achieved significantly more often in women with minor tissue loss. Perhaps the reason is that CLTI treatment differs between sexes, with women receiving significantly more minimally invasive therapies and men receiving significantly more open surgery.

Additionally, the US National Inpatient Sample database shows that women are more likely to undergo endovascular surgery for CLTI than men, and this is associated with a higher incidence of major amputation [10,36]. Therefore, in our series, it seems that endovascular treatment has been more successful than open or hybrid treatment in terms of LS in women with CLTI in the Rutherford category of 5.

In parallel, men had significantly undergone more open or hybrid surgeries for CLTI revascularization together with minor amputations and significantly achieved less LS in Rutherford category 5. It seems that in men, minor amputation as a complementary act of open or hybrid revascularization for CLTI does not give benefits in terms of LS.

Our study has limitations. Firstly, data were retrospectively collected, and some key clinical information might be missing or might not be recorded appropriately in the clinical records. The selection criteria for offering CLTI interventions are based on real-world data and are not systematic. For instance, we could not collect data on hormonal replacement/supplementation therapy for the two groups. Furthermore, we did not collect data on the extent of CLTI (for instance, below-the-knee vs. multi-level disease). Finally, the role played by unknown confounders can be relevant in an observational design. In order to mitigate potential documentation errors, patients’ data was collected from multiple hospital records, and several web meetings and phone calls were performed between the first author and the other authors during the writing of the study protocol and the patient’s recruitment period to standardize data collection.

Secondly, the timing of patient recruitment does not coincide with the actual appearance of the disease. This may cause an artificial extension, as the atherosclerosis and symptomology may have developed earlier. Thirdly, we evaluated only one year of patients treated for CLTI; as such, inter-annual variability cannot be excluded.

Finally, the staging of CLTI has not been performed using the current WIfI (Wound, Ischemia, and foot Infection) classification system. We have preferred to adopt the standard Rutherford classification, together with the presence/absence of necrosis/infection of the foot, and to exclude patients with only neuropathic (non-ischemic) diabetic foot, to make data collection easier and more realistic for the participating centers, which are mostly non-academic [37]. For the same reason, the current international suggested standards for reports dealing with risk factors and comorbidities have been adjusted or synthetized (for instance, asymptomatic carotid stenosis has not been included in CVD since in the real world not all patients treated for CLTI undergo a carotid duplex scan) [38].

## 5. Conclusions

Our observational evaluation of patients operated on for CLTI demonstrates that women are less represented and have fewer risk factors and comorbidities compared to men. However, women are struck by CLTI at an age > 75 years, a pivotal factor associated with short- and mid-term mortality, explaining why the mortality rate does not differ between the two sexes. Additionally, CRI, CAD, and tissue loss are independent negative prognostic factors for 1 year survival.

Endovascular techniques for limited tissue loss are more likely to be successful in women.

Statin therapy is an independent positive prognostic factor for 1 year survival; once again, its aggressive use in CLTI patients appears justified.

## Figures and Tables

**Figure 1 jpm-13-00316-f001:**
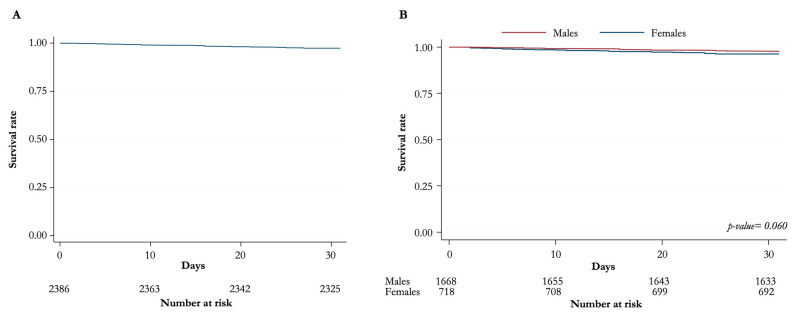
Kaplan–Meier curves for overall survival at 30 days. (**A**) Distribution of overall survival (**B**) Overall survival for males and females.

**Figure 2 jpm-13-00316-f002:**
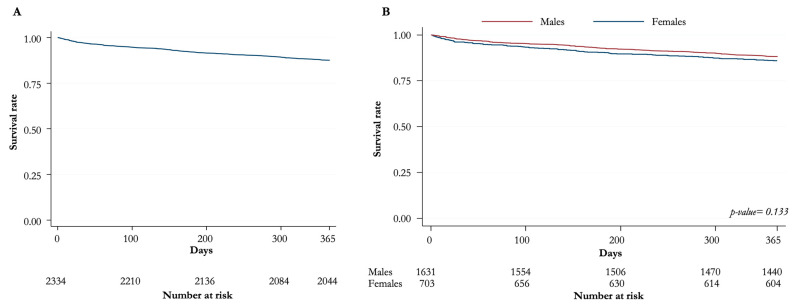
Kaplan–Meier curves for overall survival at 1 year. (**A**) Distribution of overall survival (**B**) Overall survival for males and females.

**Table 2 jpm-13-00316-t002:** Cox regression analysis to assess risk factors for 30 day mortality.

Variables		Univariate Analysis		Multivariate Analysis	
		HR (95% CI)	*p*-Value	HR (95% CI)	*p*-Value
Age > 75 yrs		3.42 (1.93–6.01)	<0.0001	3.63 (1.53–8.62)	0.003
Men		0.62 (0.38–1.03)	0.063	0.71 (0.34–1.48)	0.359
Tobacco use		1.20 (0.71–2.04)	0.501	-	-
Overt diabetes mellitus		0.80 (0.49–1.32)	0.386	-	-
Hyperlipemia:	no	Ref.	Ref.	Ref.	Ref.
	under therapy	0.60 (0.36–0.98)	0.042	0.86 (0.39–1.86)	0.693
	no therapy	0.26 (0.04–1.92)	0.187	-	-
Arterial hypertension		1.19 (0.48–2.97)	0.705	-	-
Chronic renal insufficiency:		2.01 (1.22–3.32)	0.006	1.37 (0.82–2.27)	0.227
	no	Ref.	Ref.	Ref.	Ref.
	creatinine > 2	1.78 (0.97–3.23)	0.061	0.96 (0.39–2.39)	0.937
	hemodialysis treatment	2.45 (1.25–4.81)	0.009	-	-
Coronary artery disease:		1.69 (1.03–2.78)	0.037	0.69 (0.25–1.92)	0.479
	no	Ref.	Ref.	Ref.	Ref.
	revascularized	1.79 (1.06–3.03)	0.029	2.88 (0.90–9.17)	0.074
	non-revascularized	1.40 (0.62–3.19)	0.424	-	-
Chronic obstructive pulmonary disease		1.14 (0.68–1.90)	0.624	-	-
Cerebrovascular disease		2.89 (1.54–5.41)	0.001	2.02 (0.80–5.06)	0.136
Necrosis/infection of the foot:	no	Ref.	Ref.	Ref.	Ref.
	dry	1.37 (0.64–2.95)	0.424	0.88 (0.34–2.24)	0.783
	wet	4.61 (2.56–8.31)	<0.0001	1.71 (0.73–4.02)	0.217
First intervention:	endovascular only (rarely, regenerative cellular therapy)	Ref.	Ref.	Ref.	Ref.
	any open revascularization	1.26 (0.67–2.37)	0.474	1.02 (0.48–2.17)	0.962
	any major amputation	8.96 (5.05–15.90)	<0.0001	-	-
Any intervention below the knee		0.77 (0.47–1.27)	0.309	-	-
Associated minor amputation		1.31 (0.64–2.69)	0.460	-	-
Technical success of CLTI revascularization		0.41 (0.19–0.88)	0.023	0.60 (2.25–1.47)	0.262
Postoperative antiplatelets, anticoagulants, statins:	monotherapy	Ref.	Ref.	Ref.	Ref.
	two medications	0.45 (0.27–0.77)	0.003	0.73 (0.33–1.61)	0.440
	three or more medications	0.24 (0.10–0.57)	0.001	0.39 (0.12–1.27)	0.118
Clinical success of CLTI revascularization:	worsen	Ref.	Ref.	Ref.	Ref.
	no change	0.34 (0.13–0.91)	0.032	0.50 (0.16–1.55)	0.230
	improved	0.22 (0.10–0.47)	<0.0001	0.53 (0.18–1.56)	0.248
Limb salvage		0.15 (0.09–0.25)	<0.0001	0.51 (0.19–1.36)	0.180

CLTI, chronic limb-threatening ischemia.

**Table 3 jpm-13-00316-t003:** Cox regression analysis to assess risk factors for 1 year mortality.

Variables		Univariate Analysis		Multivariate Analysis	
		HR (95% CI)	*p*-Value	HR (95% CI)	*p*-Value
Age > 75 yrs		2.50 (1.96–3.18)	<0.0001	2.14 (1.60–2.87)	<0.0001
Men		0.83 (0.65–1.06)	0.134	0.99 (0.72–1.34)	0.928
Tobacco use		0.70 (0.55–0.88)	0.002	0.89 (0.66–1.20)	0.447
Overt diabetes mellitus		0.90 (0.72–1.14)	0.395	-	-
Hyperlipemia:	no	Ref.	Ref.	Ref.	Ref.
	under therapy	0.64 (0.51–0.81)	<0.0001	0.69 (0.51–0.93)	0.015
	no therapy	0.86 (0.50–1.50)	0.602	0.97 (0.51–1.86)	0.937
Arterial hypertension		1.04 (0.69–1.55)	0.865	-	-
Chronic renal insufficiency:		1.92 (1.52–2.43)	<.0001	1.54 (1.28–1.84)	<0.0001
	no	Ref.	Ref.	Ref.	Ref.
	creatinine > 2	1.42 (1.06–1.92)	0.021	0.84 (0.59–1.21)	0.354
	hemodialysis treatment	2.89 (2.15–3.88)	<0.0001	-	-
Coronary artery disease:		1.51 (1.20–1.90)	<0.0001	1.26 (1.02–1.57)	0.036
	no	Ref.	Ref.	Ref.	Ref.
	revascularized	1.47 (1.14–1.89)	0.003	1.20 (0.88–1.65)	0.255
	non-revascularized	1.64 (1.16–2.32)	0.005	-	-
Chronic obstructive pulmonary disease		1.02 (0.80–1.29)	0.902	-	-
Cerebrovascular disease:		1.48 (1.01–2.17)	0.046	-	-
	No	Ref.	Ref.	Ref.	Ref.
	previous TIA	1.86 (1.25–2.76)	0.002	1.53 (0.95–2.46)	0.084
	previous stroke	0.39 (0.10–1.57)	0.19	0.38 (0.1–1.55)	0.177
Necrosis/infection of the foot:	no	Ref.	Ref.	Ref.	Ref.
	dry	1.83 (1.35–2.48)	<0.0001	1.42 (1.02–1.98)	0.040
	wet	3.15 (2.39–4.13)	<0.0001	2.04 (1.46–2.85)	<0.0001
First intervention:	endovascular only (rarely, regenerative cellular therapy)	Ref.	Ref.	Ref.	Ref.
	any open revascularization	0.81 (0.62–1.06)	0.122	0.86 (0.64–1.15)	0.310
	any major amputation	3.05 (2.25–4.13)	<0.0001	-	-
Any intervention below the knee		1.06 (0.84–1.33)	0.624	-	-
Associated minor amputation		1.29 (0.96–1.73)	0.091	-	-
Technical success of CLTI revascularization		0.83 (0.55–1.25)	0.37	-	-
Postoperative antiplatelets, anticoagulants, statins:	monotherapy	Ref.	Ref.	Ref.	Ref.
	two medications	0.63 (0.48–0.82)	0.001	0.93 (0.67–1.29)	0.678
	three or more medications	0.52 (0.37–0.73)	<0.0001	0.78 (0.52–1.18)	0.243
Clinical success of CLTI revascularization:	worsen	Ref.	Ref.	Ref.	Ref.
	no change	0.81 (0.50–1.34)	0.415	1.02 (0.59–1.77)	0.949
	improved	0.58 (0.38–0.88)	0.011	0.89 (0.52–1.51)	0.666
Limb salvage		0.41 (0.32–0.52)	<0.0001	0.71 (0.47–1.06)	0.095

TIA, transient ischemic attack. CLTI, chronic limb-threatening ischemia.

## Data Availability

Raw data were obtained from 25% of the Italian Divisions of Vascular Surgery and are readily available for presentation to the referees and the editors of the journal, if requested.

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
