# Peer review of "Sex-Related Differences and Factors Associated with Peri-Procedural and 1 Year Mortality in Chronic Limb-Threatening Ischemia Patients from the CLIMATE Italian Registry"

_jpm, 2023, doi:10.3390/jpm13020316_

Round 1

Reviewer 1 Report

Interesting retrospective study on short and medium term results of CLTI treatment.

This multicentric and retrospective observational study gives interesting and new informations about outcome of CLTI.

The cases number is very high due to the high number of Vascular Surgery Departments, therefore the results are to be considered significant.

Author Response

We thank the reviewer very much for the work done in revising our manuscript, and for his/her appreciation of our study. 

Reviewer 2 Report

This retrospective observational study is well designed, methods are clearly stated, statiystical analysis is adequate and the results are well reported. I have only minor comments/suggestions.

1. Have you considered correction for multiple hypothesis testing? If feasible, given the large number of tests in this study, I would suggest including a Bonferroni adjustment. If not feasible, it should be stated in the limitations.

2. Line 205/206: please provide exact thresholds applied for hyperlipemia

3. Line 425: I suggest replacing gangrene with tissue loss

Author Response

We thank you so much for your help in improving our manuscript # jpm-2116946“Sex-related differences and factors associated with peri-procedural and 1-year mortality in chronic limb-threatening ischemia patients from the CLIMATE Italian registry”.

Following our reply to the your comments.

  1. “Have you considered correction for multiple hypothesis testing? If feasible, given the large number of tests in this study, I would suggest including a Bonferroni adjustment. If not feasible, it should be stated in the limitations”.

AA: we thank the reviewer #2 for this suggestion: we did not use any Bonferroni corrections because we did not perform any multiple comparisons (i.e. post-hoc test) in our study.

  1. “Line 205/206: please provide exact thresholds applied for hyperlipemia”.

AA: we thank the reviewer #2 for this note: the cut-off value for hyperlipemia has been added.

  1. “Line 425: I suggest replacing gangrene with tissue loss"

AA: we thank the reviewer #2 for this suggestion: “gangrene” has been replaced with “tissue loss”, which is definitively more appropriate.